

# Bamboo tea: reduction of taxonomic complexity and application of DNA diagnostics based on *rbcL* and *matK* sequence data

Thomas Horn and Annette Häser

Molecular Cellbiology, Karlsruhe Institute of Technology, Karlsruhe, Germany

## ABSTRACT

**Background**. Names used in ingredient lists of food products are trivial and in their nature rarely precise. The most recent scientific interpretation of the term bamboo (*Bambusoideae*, *Poaceae*) comprises over 1,600 distinct species. In the European Union only few of these exotic species are well known sources for food ingredients (i.e., bamboo sprouts) and are thus not considered novel foods, which would require safety assessments before marketing of corresponding products. In contrast, the use of bamboo leaves and their taxonomic origin is mostly unclear. However, products containing bamboo leaves are currently marketed.

**Methods**. We analysed bamboo species and tea products containing bamboo leaves using anatomical leaf characters and DNA sequence data. To reduce taxonomic complexity associated with the term bamboo, we used a phylogenetic framework to trace the origin of DNA from commercially available bamboo leaves within the bambusoid subfamily. For authentication purposes, we introduced a simple PCR based test distinguishing genuine bamboo from other leaf components and assessed the diagnostic potential of *rbcL* and *matK* to resolve taxonomic entities within the bamboo subfamily and tribes.

**Results**. Based on anatomical and DNA data we were able to trace the taxonomic origin of bamboo leaves used in products to the genera *Phyllostachys* and *Pseudosasa* from the temperate "woody" bamboo tribe (*Arundinarieae*). Currently available *rbcL* and *matK* sequence data allow the character based diagnosis of 80% of represented bamboo genera. We detected adulteration by carnation in four of eight tea products and, after adapting our objectives, could trace the taxonomic origin of the adulterant to *Dianthus chinensis* (*Caryophyllaceae*), a well known traditional Chinese medicine with counter indications for pregnant women.

Corresponding author
Thomas Horn, thomas.horn@kit.edu

Subjects Agricultural Science, Food Science and Technology, Molecular Biology, Taxonomy, Histology
Keywords Food diagnostics, Food fraud, *Bambusoideae*, Bamboo, Consumer rights, DNA diagnostics, Herbal tea, Phylogenetics, Microscopy, PCR diagnostics

## INTRODUCTION

As consumers we rely on politics and science to counteract adverse effects of globalisation and to enforce current biological systematics as standard for food security. The European Union (EU) introduced the Novel Food Regulation (NFR) to protect consumers from

products containing unknown potentially dangerous ingredients. To market an exotic food component within the EU, business operators are required to proof that it had been consumed to a considerable amount before 1997. If the component does not comply to this criteria, it has to be considered a novel food and further steps (i.e., safety evaluations) are required before it can be marketed.

Bamboo leaf tea is considered a delicious and healthy drink in Asian countries and has found its way into the European market. The current status of bamboo leaf as food ingredient in the EU, however, is unclear. The Novel Food Catalogue (NFC) currently (June 2016) contains entries of several taxa associated with the term "bamboo": *Bambusa* spec., *Dendrocalamus latiflorus*, *D. asper*, *Gigantochloa albociliata*, *G. levis*, *Phyllostachys pubescens* and *Sinocalamus oldhamii*. Except for the first entry, all relate to the use of the stem as food source and are therefore irrelevant in regard of products using bamboo leaves. According to the entry of *Bambusa* spec. the use of leaves as food source is not known to any member state and if they were to be used as a food might be subject to the NFR and require a safety assessment. According to this information bamboo leaves used in commercial products are assumed to be derived from species of the genus *Bambusa*.

## Bamboos and bamboo tea

Bamboos are herbaceous or "woody" plants from the subfamily *Bambusoideae* (*Poaceae*) diversified in temperate and tropical Asia, South America and Africa. They are extensively used by humans (e.g., *Phyllostachys* species in China and neighbouring countries) and cultivated beyond their natural distribution range. Many species are only known from cultivation (e.g., *Bambusa* spec.). Bamboo is industrially used for construction, furniture and paper production. Domestically it is used as tool (e.g., farming, hunting, fishing, eating, weaving). The leaves of *Gelidocalamus latifolius* and *Indocalamus* species are used to wrap glutinous rice (*Wu, Raven & Hong, 2006*), those of broad-leaved species (e.g., *Sasa* species) are cut during the first 5 weeks, cleaned, dried, roasted and used for bamboo tea. For ages, bamboo tea has been considered a delicious and healthy drink in Asian countries and is now spreading to other regions (e.g., Europe). It contains neither theine nor caffeine and is rich in protein, calcium, iron, magnesium and recommended for various pharmaceutical applications, particularly stomach pain (*Liese, 2015*). In Japan the leaves of *Sasa* plants (*S. palmata*, *S. senanensis* and rarely *S. yahikoensis* and *S. kurilensis*), which are called "Kuma-zasa", have been used to treat burns or urinary hesitancy (*Sasaki et al., 2007*). In China and Indonesia, leaves of different species of *Bambusa*, *Phyllostachys*, *Fargesia* and *Indocalamus* are used for medicinal purposes (www). According to Subhuti Dharmananda (www, *Hsu et al., 1986*) the most frequently used leaves in Chinese herbal medicine are collected from the grass bamboo (*Lophatherum gracile*). It is also mentioned that leaves of the black bamboo (*Phyllostachys nigra*) and of the grass bamboo are often confused both in China and the West (*Jiao, 2003*).

Taxonomically, there are 1,641 bamboo species, 120 genera and 3 tribes (*Soreng et al., 2015*) from the subfamily *Bambusoideae* (*Poaceae*). The "woody" (lignified) bamboos are divided into two distinct tribes—the temperate (*Arundinarieae*) and tropical (*Bambuseae*) bamboos. The herbaceous bamboo tribe (*Olyreae*) is nested between them. Most of the

bamboo species mentioned in the context of leaf usage are from the temperate tribe. The grass bamboo (*Lophatherum gracile*) does not belong to the bamboo subfamily but to another *Poaceae* subfamily (*Panicoideae*), which includes members from the genus *Cymbopogon* (lemongrass), another common herbal tea ingredient. The current assumption (contained in the NFC) that bamboo leaves are derived from *Bambusa* (*Bambuseae*) species is not supported by the literature. It is more likely that the leaves are derived from a species of the temperate tribe (*Arundinarieae*).

### Herbal component diagnostics

There are different diagnostic approaches available to analyse products containing herbal components. All of them have a particular aim (i.e., to determine quality and authenticity) and are used by authorities and companies in accordance with regulations of food law. It is for example a common practise to measure concentrations of heavy metals and fungal toxins as well as to screen for foreign components (purity control). If any substance is found that is not supposed to be part of the product it is considered to be adulterated. Diagnostic approaches either use chemical components or microscopic structures that can be associated with known impurities (*Jackson & Snowden, 1990*; *Upton, Graff & Jolliffe, 2011*; *Walker & Applequist, 2012*). While the chemical approach certainly is the most direct and adequate solution when indicators are available, chemical profiles of plants are strongly influenced by environmental factors and consequently are of limited use for the determination of taxonomic (purity) authenticity. Although the morphology of plants also is highly flexible and shaped by environmental factors, a microscopic approach usually offers stable structural markers for the determination of taxonomic authenticity to a certain degree. Increasing levels of processing, however, reduce the availability of these markers and other solutions are needed.

To improve precision of the determination of taxonomic origin and to overcome the limitations of highly processed components, in recent years DNA based approaches have been developed and evaluated (*Ward et al., 2009*; *Stoeckle et al., 2011*; *Newmaster et al., 2013*; *Kazi et al., 2013*; *Coutinho Moraes et al., 2015*). While chemical approaches are restricted to a priori knowledge of contaminants and active components, morphological and DNA based approaches are able to indicate the presence of expected (genuine) or unwanted potentially harmful components (adulterants) without a priory chemical knowledge. A crucial requirement, however, are markers with appropriate diagnostic value—which also represents a priori knowledge.

Considering DNA, at least two groups of diagnostic approaches can be distinguished. Those that require to retrieve and analyse sequence information (sequencing based) and those, that operate directly on the samples DNA (probe based). While for the latter approach retrieval and analysis of sequence information also is required, after the design of a probe and the establishment of an assay, sequencing no longer is necessary.

The most prominent sequencing based approach aiming to pinpoint the taxonomic identity of a specimen is DNA barcoding (*Hebert et al., 2003*). By using a small standardized region of the genome (the barcode) and comparing the DNA sequence to a database of species barcodes, ideally, a single species name is found and by that the specimen can

be identified. Using information of DNA, this identification approach is not limited to a certain developmental stage (consider the importance of the reproductive organs of flowering plants) or particular tissue characteristics (e.g., containing starch bodies) and is not biased by environmental factors as chemical profiles are. However, DNA based approaches are far from being the ultimate answer to food related security issues. Firstly, it remains a chemically indirect approach. Secondly, DNA barcoding of land plants has been shown to be of limited use for species-level specimen identification when using the officially proposed (*CBOL Plant Working Group, 2009*) chloroplast markers (*rbcL* and *matK*). Identification success rates increase when using more variable marker regions like *psbA-trnH* (*Federici et al., 2013*), *ndhF* (*Seberg & Petersen, 2009*), the *trnL* (UAA) intron (*Taberlet et al., 2007*) or *ITS* (*Roy et al., 2010*), but frequently remain insufficient to resolve closely related species. Most DNA markers used for DNA barcoding today, have successfully been used in the past to resolve phylogenetic relationships on different hierarchical levels (*Giannasi et al., 1992*; *Chase et al., 1993*; *Gielly & Taberlet, 1994*; *Wagstaff & Olmstead, 1997*; *Frye & Kron, 2003*; *Sanchez, Schuster & Kron, 2009*). *RbcL* for example has mainly been used to elucidate higher level relationships. Although these markers have their limitations in DNA barcoding, their usefulness to answering questions of higher-level relationships remain viable.

Probe based approaches for the diagnosis of commercial products, although strongly dependent on the quality of used sequence data, are well established PCR applications to discriminate between certain taxonomic groups (*Yang et al., 2004*; *Li et al., 2007*; *Wang et al., 2007*; *Wang et al., 2010*; *Horn et al., 2012*) by detecting specific DNA regions (*Newton et al., 1989*). In contrast to approaches that use classic Sanger sequencing and require a fragment cloning step when the DNA sample is of mixed origin (i.e., samples containing DNA from different individuals or taxa, a common situation in product samples), probe based approaches can readily be applied to such samples (*Horn et al., 2013*).

When sequence information is unavailable due to the lack of primers or adequate marker regions, DNA markers can be developed using DNA fingerprinting techniques like Random Amplified Polymorphic DNA (RAPD) or Amplified Fragment Length Polymorphism (AFLP) (*Wang et al., 2001*; *Lee et al., 2006*; *Dnyaneshwar et al., 2006*; *Huh & Bang, 2006*; *Marieschi et al., 2010*; *Torelli, Marieschi & Bruni, 2014*).

## Aim

Due to information discrepancies between the Novel Food Catalogue and available literature about the use of bamboo, our primary objective was to reduce taxonomic complexity of the term "bamboo" in context of food products. To achieve this, we aimed to determine the taxonomic origin of bamboo leaves used in corresponding products. In detail, we wanted to answer two questions: 1. Do bamboo tea leaves originate from the temperate or tropical "woody" bamboos; and 2. Are we able to determine the genus of origin?

We also aimed to provide preliminary product diagnostics using microscopic features as well as a PCR based assay. To maximize diagnostic information content we specify the general diagnostic value of corresponding DNA markers using character based DNA

**Table 1** **Analyzed product accessions (Acc). Three bamboo teas (BT) and five fruit teas (FT) are listed.** While bamboo tea only contains one leaf component, the fruit teas are mixtures of fruit fragments and one (i.e., bamboo) or two (i.e., bamboo and lemongrass) leaf components. GenBank sequence accession numbers of *rbcL*a, *rbcL*b, *matK-KIM* and *ITS* generated in the current study are also included.

| Acc | Type | Leaf component(s) | rbcLa | rbcLb | matK-KIM | *ITS* |
|-----|------|-------------------|-------|-------|----------|-------|
| P1 | FT | Bamboo, lemongrass | KU722894 | KU722852 | KU722866 | KU722880 |
| P2 | FT | Bamboo | KU722893 | KU722851 | KU722865 | KU722879 |
| P3 | FT | Bamboo | KU722891 | KU722849 | KU722863 | KU722877 |
| P4 | BT | Bamboo whole leaf | KU722892 | KU722850 | KU722864 | KU722878 |
| P5[a] | BT | Bamboo | KX233507 | KX233494 | KX233503 | – |
| P6 | FT | Bamboo | KX233506 | KX233493 | KX233502 | – |
| P7 | FT | Bamboo, lemongrass | KX233505 | KX233492 | KX233501 | – |
| P8[a] | BT | Bamboo | KX233508 | KX233495 | KX233504 | – |

**Notes.**
[a] Fine fragmented leaf material is contained in tea bags.

diagnostics. In general, we aimed to use available information to resolve any accessible taxonomic entity, not limiting our efforts to species specificity.

After discovering the adulteration of corresponding tea products with carnation, we naturally extended our objectives by including the adulterant in all analyses.

## MATERIAL AND METHODS

### Commercial samples and reference specimens

We acquired three bamboo teas and five fruit teas containing bamboo leaves from eight different companies (Table 1). According to the ingredient list, all of the fruit teas contained bamboo as leaf component, and two of them additionally lemongrass. As references for potential sources of bamboo tea leaves, we acquired specimens of 13 bambusoid species (Table 2) and cultivated them in the botanic garden of the Karlsruhe Institute of Technology. To be able to analyse all leaf components of corresponding teas, we included two lemongrass specimens from the botanic garden collection into our analysis. Finally, after the adulteration of teas with a *Dianthus* species became clear, we also included specimens of eight *Dianthus* species (Table S2).

The authors identified bamboo (*Bambusoideae*) and lemongrass (*Cymbopogon*) specimens to genus level (*Farrelly, 1984*; *Wu, Raven & Hong, 2006*; *Wu, Raven & Hong, 2007*). In case of *Dianthus*, at least one specimen of each species was identified to species level (*Wu, Raven & Hong, 2001*; *Jäger et al., 2008*).

Specimen information (i.e., details, images and sequences) are available through the Barcoding of Life Data Systems (BOLD) website. Note that you need to be logged in to BOLD to access the data.

### Morphological and anatomical evaluations

Small rectangle hand-sections were made in the centre and at the margin of the leaf-blades of the first fully developed dried leaves of reference plants. Leaf fragments were isolated from all commercial products. After visual inspection of specimens using a stereo microscope (Leica S6D), the adaxial and abaxial leaf surfaces were brightened with 60%

**Table 2  Bamboo specimens.** Accessions (Acc) B1–B13 of *Bambusoideae* species, their taxon names (according to source) and GenBank sequence accession numbers of three plastid DNA regions (*rbcLa*, *rbcLb* and *matK-KIM*) generated in the current study are listed.

| Acc | Taxon | rbcLa | rbcLb | matK-KIM |
|---|---|---|---|---|
| **Bambuseae** | | | | |
| B1 | *Bambusa multiplex* | KX146450 | KX146413 | KX146427 |
| B2 | *Dendrocalamus giganteus* | KX146452 | KX146415 | KX146429 |
| **Arundinarieae** | | | | |
| B3 | *Phyllostachys aureosulcata* | KX146453 | KX146416 | KX146430 |
| B4 | *Phyllostachys edulis* | KX146454 | KX146417 | KX146431 |
| B5 | *Phyllostachys nigra* | KX146455 | KX146418 | KX146432 |
| B6 | *Phyllostachys violascens* | KX146456 | KX146419 | KX146433 |
| B7 | *Pseudosasa japonica* | KX146457 | KX146420 | KX146434 |
| B8 | *Sasa borealis* | KX146458 | KX146421 | KX146435 |
| B9 | *Sasa kurilensis* | KX146459 | KX146422 | KX146436 |
| B10 | *Sasa palmata* | KX146460 | KX146423 | KX146437 |
| B11 | *Sasa veitchii* | KX146461 | KX146424 | KX146438 |
| B12 | *Semiarundinaria fastuosa* | KX146462 | KX146425 | KX146439 |
| B13 | *Thamnocalamus tessellatus*[a] | KX146451 | KX146414 | KX146428 |

**Notes.**

[a] Has recently been assigned to a separate genus: *Bergbambos* (*Stapleton, 2013*).

chloral hydrate (Carl Roth GmbH) and analysed using a light microscope (Leica DM750). Both instruments are equipped with a digital image system (Leica EC3) that was used to document macroscopic and microscopic leaf structures.

## DNA based evaluations

For DNA based evaluations, we chose to retrieve sequence information from the ribulose-bisphosphate carboxylase oxygenase large subunit (*rbcL*) employing primers for *rbcLa* (*rbcL* pos. 1–599) (*Soltis, Soltis & Smiley, 1992*; *Kress et al., 2009*) and *rbcLb* (*rbcL* pos. 315–1,175) (*Dong et al., 2014*), maturase K (*matK*) employing primers for *matK-KIM* (Ki-Joong Kim, unpublished) and the Internal Transcribed Spacers (*ITS*) of nuclear ribosomal DNA employing primers ITS5 and ITS4 (*White et al., 1990*).

**DNA isolation:** DNA was isolated from sterilized leaf samples of bamboo (Table 2), lemongrass and *Dianthus* (Table S2) specimens as well as from product samples (Table 1), using the innuPREP Plant DNA Kit (Analytik Jena AG) following the vendor's instructions. Product samples consisted of particular leaf fragments that were selected from commercial products with the support of a stereo microscope (Leica S6D). Products containing more than one leaf component (i.e., bamboo and lemongrass) were sampled twice: a leaf fragment of the assumed bamboo component for PCR and sequencing and a fragment mixture containing both components for PCR diagnostics. Purity and concentration of isolated DNA was determined using a spectrophotometer (Nanodrop, Thermo Fisher Scientific, Germany).

**Amplification and sequencing:** A 30 µL reaction volume containing 20.5 µL nuclease free water (Lonza, Biozym Scientific GmbH), 1-fold Thermopol Buffer from New England Biolabs GmbH (NEB), 1 mg/ml bovine serum albumin, 200 $\mu$mol dm$^{-3}$ dNTPs (NEB), 0.2 $\mu$mol dm$^{-3}$ of forward and reverse primer (Table S1), 100–150 ng DNA template and 3 units of Taq polymerase (NEB) was used to amplify marker sequences. The PCR program consisted of an initial denaturation step at 95 °C C for 1 min, 35 cycles of 30 s denaturation at 94 °C, 30 s annealing with primer specific temperatures (Table S2) and 1 min extension at 68 °C, concluding with a 10 min final extension step at 68 °C. The PCR reaction was subsequently evaluated by agarose gel electrophoresis (AGE) using NEEO ultra-quality agarose (Carl Roth GmbH). DNA was visualized using SYBR Safe (Invitrogen, Thermo Fisher Scientific Germany) and subsequent blue light excitation. The fragment size was determined using a 100 bp size standard (NEB). Amplified DNA was purified using a NucleoSpin Gel and PCR Clean-up kit (MACHEREY-NAGEL GmbH). Sequencing was outsourced to Macrogen Europe (Netherlands).

**Sequence assembly and BLAST analysis of product samples**: Sequencing results were assembled using a perl script. Raw data was converted to fasta (phred 20) and bi-directional reads merged to recover ambiguous characters (i.e., N). For additional quality control IUPAC consensus sequences were generated and inspected. Resulting sequences of product samples were used in a Basic Local Alignment Search Tool (BLAST) analysis to approximate taxonomic identity.

**Preparation of marker datasets**: For the phylogenetic analysis and DNA diagnostics evaluation, we combined specimen sequences with sequences of relevant taxonomic groups retrieved from GenBank (Tables S3 and S4). Sequence collections of coding regions (i.e., *rbcL* and *matK*) were aligned using MUSCLE (*Edgar, 2004a*; *Edgar, 2004b*) and the *ITS1-5.8S-ITS2* region using MAFFT with L-INS-i option (*Katoh, 2002*; *Katoh & Standley, 2013*). Subsequently, primer sites were removed and alignments trimmed using sequences of our specimens. Finally, each dataset was evaluated for its information content (alignment length, variable positions, parsimony information and singleton sites).

## Phylogenetic diagnostics

To reduce taxonomic complexity of the commercially used term "bamboo," thus determining the taxonomic origin of bamboo tea leaves, trees were computed using Unweighted Pair Group Method with Arithmetic Mean (UPGMA), Maximum Parsimony (MP) and Maximum Likelihood (ML) algorithms implemented in MEGA6 (*Tamura et al., 2013*). For UPGMA the evolutionary distances were computed using the p-distance method (*Nei & Kumar, 2000*) with all ambiguous positions removed for each sequence pair. The MP tree was obtained considering all sites using the Subtree-Pruning-Regrafting (SPR) algorithm (*Nei & Kumar, 2000*) with search level 1 in which the initial trees were obtained by the random addition of sequences (10 replicates). The evolutionary history inferred by using the ML method was based on substitution models in combination with evolutionary rate differences among sites that had the lowest BIC (Bayesian Information Criterion) scores determined by analysing each dataset using MEGA6. Details are summarized in Table 3.

**Table 3** Substitution models (K2, Kimura 2-parameter; T92, Tamura 3-parameter; GTR, General Time Reversible) and evolutionary rates among sites (+G, discrete gamma distribution) used for ML analysis.

| Dataset | Model | Rates | BIC |
|---|---|---|---|
| *rbcLa* | K2 | +G | 5,217 |
| *rbcLb* | T92 | +G | 7,177 |
| *rbcL* | T92 | +G | 9,380 |
| *matK-KIM* | GTR | +G | 10,829 |
| *r+m* | GTR | +G | 19,223 |

Initial tree(s) for the heuristic search were obtained by applying the Neighbor-Joining (NJ) method to a matrix of pairwise distances estimated using the Maximum Composite Likelihood (MCL) approach. All trees were bootstrapped (*Felsenstein, 1985*) using 500 replicates. Additionally, we computed UPGMA, MP and ML trees using concatenated datasets (*rbcL = rbcLa* and *rbcLb*; *r+m = rbcLa*, *rbcLb* and *matK-KIM*). The results were analysed by first collapsing branches corresponding to partitions reproduced in less than 50% bootstrap replicates and recording bootstrap support values for relevant monophyletic groups (sensu *Soreng et al. (2015)*) for each dataset and each of the used phylogenetic approaches. In detail, we focused on support for the bambusoid subfamily (*Bambusoideae*), the three tribes (*Arundinarieae*, *Bambuseae* and *Olyreae*) and any sub-tribal entity that would offer sufficient support. All datasets and trees have been deposited in TreeBase . For representation, the dataset and algorithm providing most support for corresponding monophyletic groups was edited using FigTree V1.4.2 (*Rambaut, 2014*).

### Character based diagnostics

The PCR diagnostics approach had two objectives. Firstly, we aimed to establish a simple but efficient PCR based differentiation between bamboo (= subfamily = *Bambusoideae*), a common tea leaf component (lemongrass = genus = *Cymbopogon*) from the same family (*Poaceae*) and the adulterant from the genus *Dianthus* (*Caryophyllaceae*). Secondly, we wanted to assess the diagnostic potential of the used marker regions to resolve entities within the bambusoid subfamily and the adulterant genus.

**Differentiation of bamboo tea components and adulterant**: For the PCR based differentiation protocol we chose *rbcLa* which had been successfully used before to differentiate above the generic level (*Horn et al., 2012*; *Horn et al., 2013*). Using the *rbcLa* dataset, we designed primers to detect single nucleotide polymorphisms (*Newton et al., 1989*; *Ward et al., 2009*) characteristic for bamboo, lemongrass and carnation. Nucleotide differences between the mentioned components were determined and potential diagnostic primer sequences extracted. One suitable primer for each group was chosen and destabilized according to *Newton et al. (1989)* (Table S1). The theoretical suitability of a diagnostic primer was determined using primer3 (*Untergasser et al., 2007*; *Untergasser et al., 2012*) with default settings. The diagnostic primers were evaluated in a multiplex PCR with the universal primer-pair (*rbcLa*). For each diagnostic primer a separate set of 10 µL PCR reactions containing 6.5 µL nuclease free water (Lonza; Biozym Scientific GmbH), 1-fold

Thermopol Buffer (NEB), 1 mg/ml bovine serum albumin, 200 $\mu$mol dm$^{-3}$ dNTPs (NEB), 0.3 $\mu$mol dm$^{-3}$ of universal forward primer, 0.2 $\mu$mol dm$^{-3}$ of universal and diagnostic reverse primer, 25–50 ng DNA template and 0.5 units of Taq polymerase (NEB) was used. We used the same PCR program as employed for amplification of the *rbcLa* marker region (see above). The PCR products were evaluated by gel electrophoresis using high resolution agarose (Carl Roth GmbH).

**Assessment of diagnostic potential**: To assess the diagnostic potential of DNA markers, we used a character bases DNA barcoding appraoch: Barcoding with LOGic (*Weitschek et al., 2013*; *Bertolazzi, Felici & Weitschek, 2009*). We prepared separate single and multi-locus datasets containing only sequences of *Bambusoideae* and *Caryophyllaceae* respectively. Sequences were labelled according to specific taxonomic classes. For the *Bambusoideae* dataset we tested tribe and genus as diagnostic entities. For *Dianthus* we only tested species as diagnostic entity. Since in *Dianthus* the general evaluation showed limited variation within *rbcL*, we chose to evaluate only *matK-KIM* as cytoplasmic marker. Additionally, we included an *ITS* dataset that contained all available *Dianthus* GenBank sequences, regardless if data also existed for the cytoplasmic markers. The BLOG algorithm was subsequently used with standard settings (except padding = 1, percslicing = 100 and exclusivefs = 1) to find characters or character combinations (LOGic formulas) that are representative for a class, thus can be used to diagnose product samples.

# RESULTS

## Anatomical evaluation

Morphology as the study of forms visible to the unaided eye, in food diagnostics is complemented by anatomy, the study of cellular structures. For an intermediate between morphology and anatomy, in this study we used the term "macroscopic." The magnification used does not yet allow to observe cellular structures in detail, but eases the study of their morphological manifestations. Both, microscopic and macroscopic anatomy are common techniques used in food diagnostics (*Hohmann & Gassner, 2007*).

**Macroscopic features**: A characteristic of the bambusoid leaf is a mosaic pattern of longitudinal and transverse veinlets, so called tesselation. The evaluation of leaf samples taken from our bamboo specimens supports the description of *Farrelly (1984)* wherein tesselation of the leaf is a visible characteristic of hardy, monopodial species (*Arundinarieae*, Fig. 1E) and is hidden from the unaided eye by tissue in sympodial bamboos whose leaves are often more tough and leathery (*Bambuseae*, Fig. 1F). Evaluating the leaf samples taken from herbal tea products, tesselation was observed in samples P5–P8 (e.g., Figs. 1C and 1D). While leaf fragments with tesselation always were fragments in longitudinal and transversal respect, leaf components of the remaining products P1–P4 consisted of thin (approximately 4 mm) linear to lanceolate leaves (Fig. 1B), in some instances oppositely arranged at the fragment of a shoot. The observed arrangement of leaves is in direct conflict with the index of contents of corresponding products clearly stating "bamboo leaves" as component. The leaves of *Poaceae* plants, however, are usually alternately arranged at

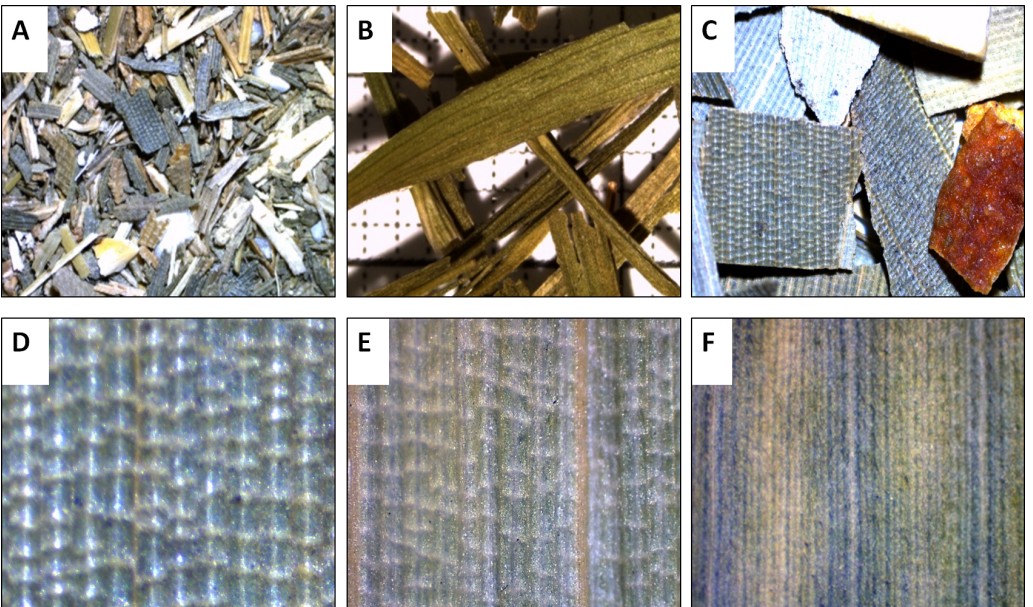

**Figure 1** **Macroscopic features of bamboo tea products (A–D) and bamboo leaf samples (E and F).** Leaf fragments (10×) of bamboo tea bag product (A), adulterant component (B) and of a bamboo fruit tea (C). Leaf surface (adaxial, 40×) of bamboo tea component (D) in comparison to *Arundinarieae* (E, *Sasa palmata*) and *Bambuseae* (F, *Bambusa multiplex*) dried leaf samples.

the shoot (*Wu, Raven & Hong, 2006*). Since bamboo tea is also available in a form where components are so small, that the arrangement of leaves cannot be determined (tea bags), microscopic features need to be considered.

**Microscopic features:** Using light microscopy (100 x), tesselation was observed in all bamboo specimens. Additionally, characteristic structures of the bambusoid leaf (*Wu, 1962*; *Vieira et al., 2002*) were observed: epidermal cells—longitudinal bands composed of long rectangular cells with wavy lateral walls and alternating short rectangular cells, separated by bulliform cells (*Beal, 1886*; *Alvarez, Rocha & Machado, 2008*) in the upper epidermis (Fig. 2A); and modified epidermal cells—stomata of the *Poaceae* type, microhairs, spines, papillae, bristles and silica cells (Figs. 2B–2D). The microscopic evaluation of commercial samples P5–P8 was congruent with the results from bamboo specimens, showing bambusoid features (e.g., tesselation: Fig. 2E). Samples P1–P4 did not display any bambusoid characteristics but stomata of a different type than *Poaceae* (Figs. 2H and 2I) and crystal druses (Fig. 2J) along main veins and in intercostal regions. We recognised anomocytic stomata common in *Caryophyllaceae* and *Ranunculaceae* (*Rohweder, Schlumpf & Krattinger, 1971*) predominated by the diacytic form. This suggests that samples P1–P4 probably originated from a *Caryophyllaceae* plant.

## DNA based evaluation

All three cytoplasmic markers were retrieved with great success regarding PCR and sequencing results. *ITS* however turned out to be particularly problematic with bamboo samples and could only be retrieved for *Dianthus* specimens and product samples P1–P4.

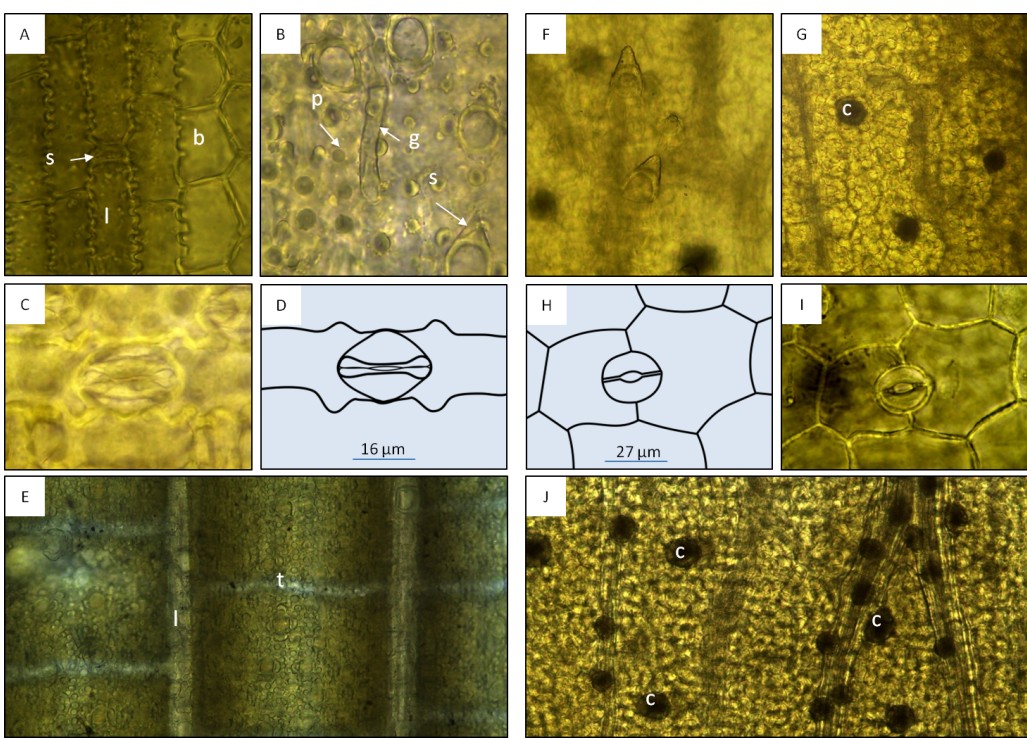

**Figure 2  Macroscopic features of bamboo tea products (A–D) and bamboo leaf samples (E and F).** Microscopic features of the bambusoid leaf observed in specimens (A–D, 400×) and product samples (E, 100×), and microscopic features of *Dianthus chinensis* observed in specimens (F and G, 100×; H and I, 400×) and product samples (J, 100×). (A) Adaxial epidermis of *Bambusa multiplex* showing longitudinal bands of long rectangular cells (l) with wavy lateral walls and alternating short rectangular cells (s) separated by bulliform cells (b). (B) Abaxial modified epidermal structures of *Phyllostachys edulis* (p, papillae; g, geniculate hair; s, spine). (C and D) Abaxial epidermis with *Poaceae* type stomata of *Sasa palmata*. (E) Epidermis with longitudinal (l) and transverse veinlets (tesselation) observed in product samples. (F) Leaf epidermis of *D. chinensis* showing unicellular trichomes. (G) Mesophyll of *D. chinensis* showing crystal druses (c). H and I: Abaxial epidermis of *D. chinensis* with anomocytic stomata (here diacytic). (J) Mesophyll with crystal druses (c) along main veins and in intercostal regions observed in product samples.

Preferential and co-amplification of *ITS* from fungal trace DNA prevented the retrieval of a complete dataset for bamboo specimens. Similar problems have been reported by *Zhang, Wendel & Clark (1997)*.

## General assessments

**BLAST analysis of product DNA sequences**: Single locus markers (*rbcLa*, *rbcLb* and *matK-KIM*) were used in a BLAST analysis. Two groups could be distinguished: P1–P4 returned hits indicating close relation to *Dianthus* (*Caryophyllaceae*) and P5–P8 returned hits belonging to genera of *Bambusoideae*.

   **Information content:** Final single marker dataset alignments contained 553, 814 and 837 nucleotides for *rbcLa*, *rbcLb* and *matK-KIM* respectively. Combining *rbcLa* and *rbcLb* (*rbcL*) excluding redundant data, the alignment had 1,126 positions. The combination of *rbcLa*, *rbcLb* and *matK-KIM* had 1,963 positions respectively. The *ITS* dataset derived from our *Dianthus* specimens contained 611 nucleotides. After including GenBank accessions

**Table 4** **Information content of bamboo and *Dianthus* genetic marker datasets comprised of 43 and 14 DNA sequences respectively.** Sequences were obtained from life specimens, product samples and GenBank. Length (Len), conserved (Con), variable (Var), parsimony informative (PaI) and singleton (Sin) characters as well as the number of haplotypes (Hap) are listed for cytoplasmic markers (*rbcLa*, *rbcLb* and *matK-KIM*) and combinations (*rbcL* = *rbcLa* + *rbcLb* and *r+m* = *rbcLa* + *rbcLb* + *matK-KIM*). For *Dianthus* the same information is listed for a nuclear (*ITS*) marker: one simple dataset only containing sequences of species also present in the cytoplasmic datasets and one extended dataset ([a]) including all available ITS GenBank sequences of *Dianthus*.

| Marker | Len | Con | Var | % | PaI | % | Sin | % | Hap |
|--------|-----|-----|-----|-----|-----|-----|-----|-----|-----|
| | | | | **Bamboo** | | | | | |
| *rbcLa* | 553 | 523 | 30 | 5.4 | 17 | 56.7 | 13 | 43.3 | 17 |
| *rbcLb* | 814 | 761 | 53 | 6.5 | 26 | 49.1 | 27 | 50.9 | 13 |
| *rbcL* | 1,126 | 1,057 | 69 | 6.1 | 35 | 50.7 | 34 | 49.3 | 22 |
| *matK* | 837 | 748 | 89 | 10.6 | 40 | 44.9 | 49 | 55.1 | 25 |
| *r+m* | 1963 | 1805 | 158 | 8.0 | 75 | 47.5 | 83 | 52.5 | 31 |
| | | | | ***Dianthus*** | | | | | |
| *rbcLa* | 553 | 553 | 0 | 0.0 | 0 | 0.0 | 0 | 0.0 | 1 |
| *rbcLb* | 814 | 808 | 6 | 0.7 | 6 | 100.0 | 0 | 0.0 | 3 |
| *rbcL* | 1,126 | 1,120 | 6 | 0.5 | 6 | 100.0 | 0 | 0.0 | 7 |
| *matK* | 837 | 826 | 11 | 1.3 | 8 | 72.7 | 3 | 27.3 | 3 |
| *r+m* | 1,963 | 1,946 | 17 | 0.9 | 14 | 82.4 | 3 | 17.6 | 7 |
| *ITS* | 611 | 566 | 45 | 7.4 | 39 | 86.7 | 6 | 13.3 | 12 |
| *ITS*[a] | 618 | 498 | 120 | 19.4 | 72 | 60.0 | 48 | 40.0 | 87 |

**Notes.**
[a] Extended dataset.

(Table S4) the dataset comprised 85 sequences with 618 positions. Information content (i.e., number and proportion of variable sites and parsimony informative positions) within *Bambusoideae* and *Dianthus* datasets is shown in Table 4. In both taxonomic groups most variation among single locus cytoplasmic markers was detected in the *matK-KIM* region. Considering parsimony information, *rbcLa* in bamboo and *rbcLb* in *Dianthus* show the highest proportion (57 and 100% respectively). The combination of single locus data obviously contains all variation and informative sites but reduces the proportion in combined datasets. Among the *Dianthus* datasets the nuclear marker (*ITS*) contains the highest variation and thus delivers most information. Sequence data generated in this study are deposited in BOLD and GenBank. Sequence accessions from other studies that were included in this study are contained within Tables S3 and S4.

### Phylogenetic diagnostics

**Clade support**: Comparing the support for relevant clades using different phylogenetic methods with single and multi-locus datasets reveals several interesting aspects (Fig. 3).

Sequence accessions of *Borinda* (*Arundinarieae*) and *Chusquea* (*Bambuseae*) cluster in the *Bambuseae* and *Arundinarieae* clade respectively, invalidating the monophyly of this clades. We therefore introduced additional evaluation classes: *Arundinarieae* modified (Fig. 3-4, line graph) and *Bambuseae* modified (Fig. 3-2, line graph). For these classes the position of both mentioned sequence accessions was ignored when assessing monophyly.

 

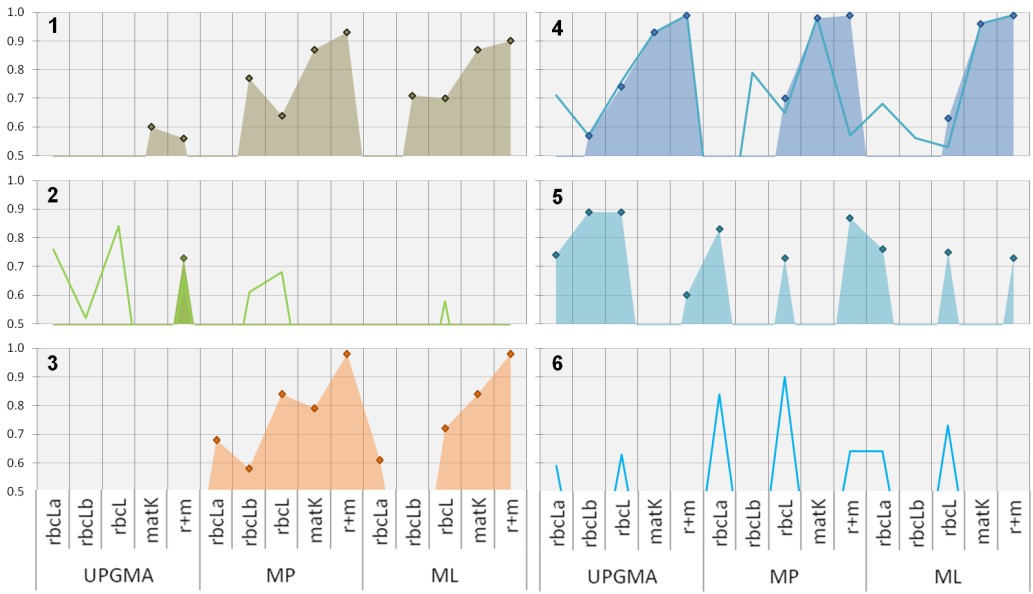

**Figure 3  Phylogenetic evaluation of bambusoid clades.** The figure shows statistical support (*y*-axis, percent bootstrap replicates in decimal form) for relevant clades (1: *Bambusoideae*, 2: *Bambuseae*, 3: *Olyreae*, 4: *Arundinarieae* , 5: *Sasa*, 6: *Phyllostachys*) using UPGMA, MP and ML methodology with single locus (*rbcLa*, *rbcLb* and *matK-KIM*) and multi-locus (*rbcL*: *rbcLa + rbcLb*; *r+m*: *rbcL + matK-KIM*) data. While diamonds and filled areas indicate statistical support >50% for strict clade composition, lines represent support for alternatives (refer to the results section for further details).

The bamboo subfamily (*Bambusoideae*, Fig. 3-1 ) is supported with more than 50% of replicates by all marker regions except *rbcLa*, using MP and ML methodology. When using *rbcLa* the *Oryzoideae* clade resides among the bamboo members making *Bambusoideae* a non-monophyletic clade. Support for the bambusoid subfamily is constantly equal or above 70% except when using the combined *rbcL*-sub-regions and the MP approach. With support between 50 and 60% of two of five tested datasets (*matK* and *r+m*), UPGMA only gives weak and inconsistent support for the subfamily.

Focusing on the three bambusoid tribes (*Arundinarieae*, Fig. 3-4; *Bambuseae*, Fig. 3-2; *Olyreae*, Fig. 3-3) none of the markers and methods strongly supports all corresponding clades at the same time. Applying MP with *matK* or combined cytoplasmic data yields high support (>70%) for *Arundinarieae* and *Olyreae*. Both clades are also supported according to ML, using combined *rbcL* (>63%), *matK* (>84%) and combined cytoplasmic (>98%) datasets. The *Olyreae* clade (Fig. 3-3) receives consistent support using any dataset with the MP approach (*rbcLb* 58%–*r+m* 98%). Similarly, except when using the *rbcLb* dataset, the ML approach offers high support (*rbcLa* 61%–*r+m* 98%). The *Arundinarieae* clade (Fig. 3-4) also is consistently supported by all three phylogenetic approaches, particularly when using *matK* (UPGMA 93%–MP 98%) or the combined cytoplasmic dataset (99%). Considering an alternative taxonomic configuration (*Arundinarieae* mod., line in Fig. 3-4) some of the single datasets offer support for the corresponding clade. However, a significant difference between the support for the *Arundinarieae* clade (99%) and the modified clade (57%) can be observed when using the combined cytoplasmic dataset in a MP analysis.

The *Bambuseae* clade (Fig. 3-2) only once is supported above 50% (UPGMA: *r+m*) unless considering an alternative taxonomic configuration (*Bambuseae* mod., line in Fig. 3-2). In all cases where a *Bambuseae* mod. clade is supported, the *Chusquea* sequence accession fails to cluster with other *Bambuseae* sequences. In every other instance where general support for *Bambuseae* is missing, the *Chusquea* sequence clusters with *Sasa* (MP, ML: *rbcL*b) and only some of the *Bambusaea* sequences form supported clusters. A sister clade consisting of *Otatea* and *Olmeca* is consistently formed (UPGMA: *matK*; MP: *rbcL*a, *matK*, *r+m*; ML: *rbcL*a, *rbcL*b, *r+m*) along other *Bambuseae* sequences. In the ML analysis using *matK* the *Olyrae* clade resides within the *Bambuseae* clade resulting in the non-monophyly of the latter.

Support on the genus level is rare. Only *Sasa* (Fig. 3-5 ) and *Thamnocalamus* (both *Arundinarieae*) form monophyletic clades. The *Sasa* clade can be observed in 10 of 15 cases, all based on *rbcL* data. A monophyletic *Thamnocalamus* clade can only be observed when using *rbcL*a data. Since product samples frequently clustered within a clade containing *Phyllostachys* (*Arundinarieae*) we introduced another evaluation class: *Phyllostachys* modified (Fig. 3-6, line graph). This class consists of all *Phyllostachys*, *Fargesia*, *Indocalamus* and *Drepanostachyum* sequence accessions. The modified clade can be observed using *rbcL*a and the combined *rbcL* dataset (UPGMA, MP and ML) as well as when using the combined cytoplasmic dataset (MP). Also in this case, support appears to be solely derived from *rbcL* data. Although *rbcL*b data does not offer direct support, its contribution to the combined dataset can clearly be observed by increased support values (e.g., up to almost 10% in ML analysis).

All other *Poaceae* groups (i.e., *Bambusoideae* outgroups *Oryzoideae* and *Pooideae*, and secondary component group *Panicoideae*) receive consistent and strong (>85%) support (Fig. 4). One exception worth mentioning is the low (MP: 52%) and missing support (UPGMA and ML) for *Panicoideae* (represented by *Cymbopogon* and *Lophaterum*) when using *rbcL*b data (not shown).

Support for the genus of the adulterant (*Dianthus*, >= 72%) as well as the corresponding family (*Caryophyllaceae*, 100%) and outgroup (*Silene*, >= 64%) is consistent and strong (Fig. 4) with rare low points (not shown), i.e., using *matK* data with ML (*Silene*) and using *rbcL*b data with ML (*Dianthus*).

**Phylogenetic representation**: Using the combined cytoplasmic dataset with sequences recovered from product components and building a MP tree, basically visualizes the BLAST results within an evaluated phylogenetic framework (Fig. 4). Product samples P1–P4 clearly are located within the *Dianthus* (*Caryophyllaceae*) clade and product samples P5–P8 are located within the *Arundinarieae* (*Bambusoideae*, *Poaceae*) clade.

### Diagnostic analysis

**Differentiation of tea components and adulterant**: Based on a *rbcL*a dataset containing bamboo, lemongrass and *Dianthus* sequences we designed three reverse ARMS primer (Table S1) with diagnostic nucleotides located at position 407, 254 and 223 respectively. The evaluation of multiplex PCRs, applying these specific primers in separate reactions together with *rbcL*a universal primers (Fig. 5), shows sufficient specificity and amplification
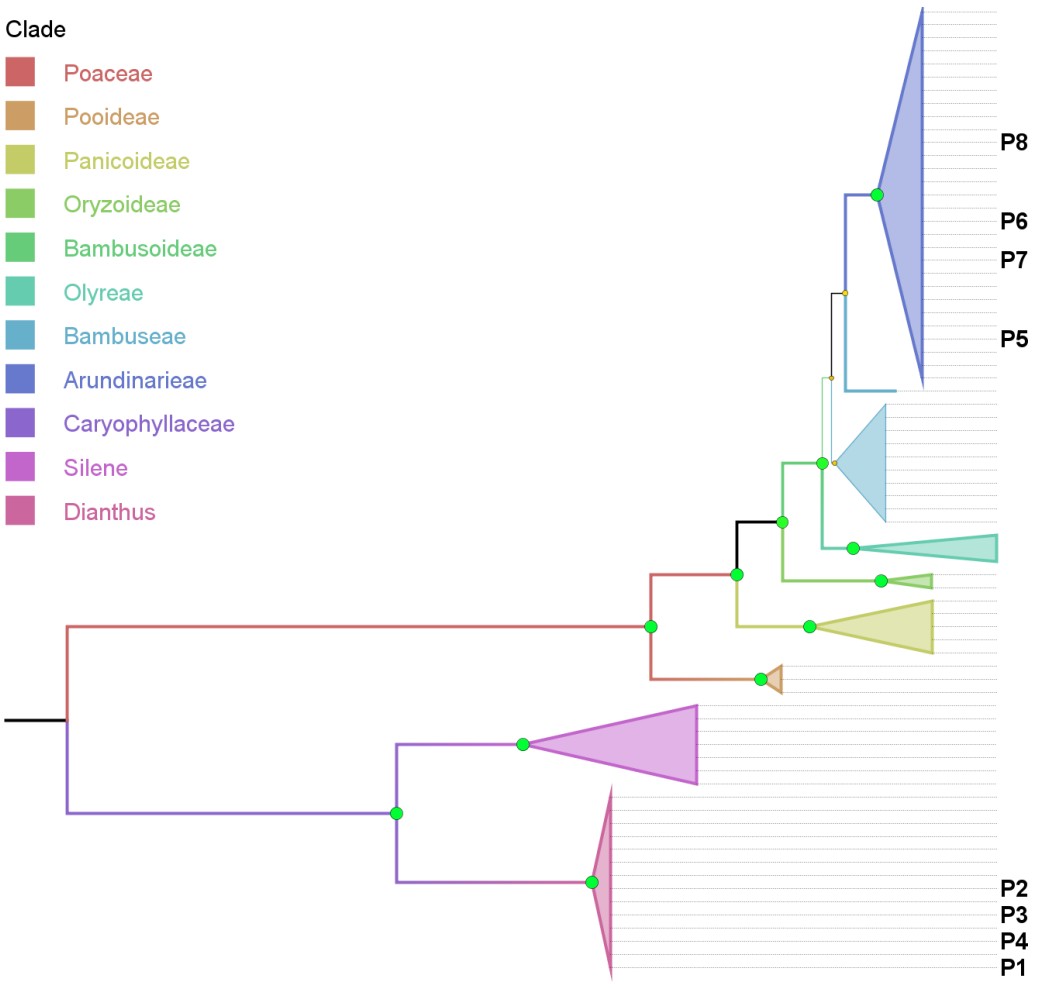

**Clade**
- Poaceae
- Pooideae
- Panicoideae
- Oryzoideae
- Bambusoideae
- Olyreae
- Bambuseae
- Arundinarieae
- Caryophyllaceae
- Silene
- Dianthus

**Figure 4  Phylogenetic tree based on combined cytoplasmic sequence data using Maximum Parsimony (MP).** The percentage of replicate trees in which the associated taxa clustered together in the bootstrap test (500 replicates) are indicated by the size and colour (green-yellow-red = 100-50-0%) of the nodes. The analysis involved 74 nucleotide sequences and 1999 positions in the final dataset. P1–P8 indicates the location of sequence data retrieved from product samples.

of diagnostic fragments (bamboo, 457 bp; lemongrass 306 bp; *Dianthus* 268 bp) to differentiate the three leaf components present in commercial tea products. Products P1–P4 show diagnostic fragments of size 268 bp indicating the presence of *Dianthus* (Fig. 5D) and are lacking bamboo diagnostic fragments (Fig. 5B). Products P5–P8 show the exact opposite pattern, no diagnostic fragments specific for *Dianthus* but for bamboo. Additionally, the presence of lemongrass in products P1 and P7 is shown by diagnostic fragments of the corresponding size (Fig. 5L, 306 bp). All specimens of the corresponding groups have been tested for positive reaction using the diagnostic primer and negative (null) reaction using any diagnostic primer of different groups.

**Assessment of diagnostic potential**: The evaluation of bambusoid tribe diagnostics using BLOG shows consistency among markers (Table S6). Only the *Arundinarieae* tribe shows 4% false negative classifications when using the *rbcLa* dataset.
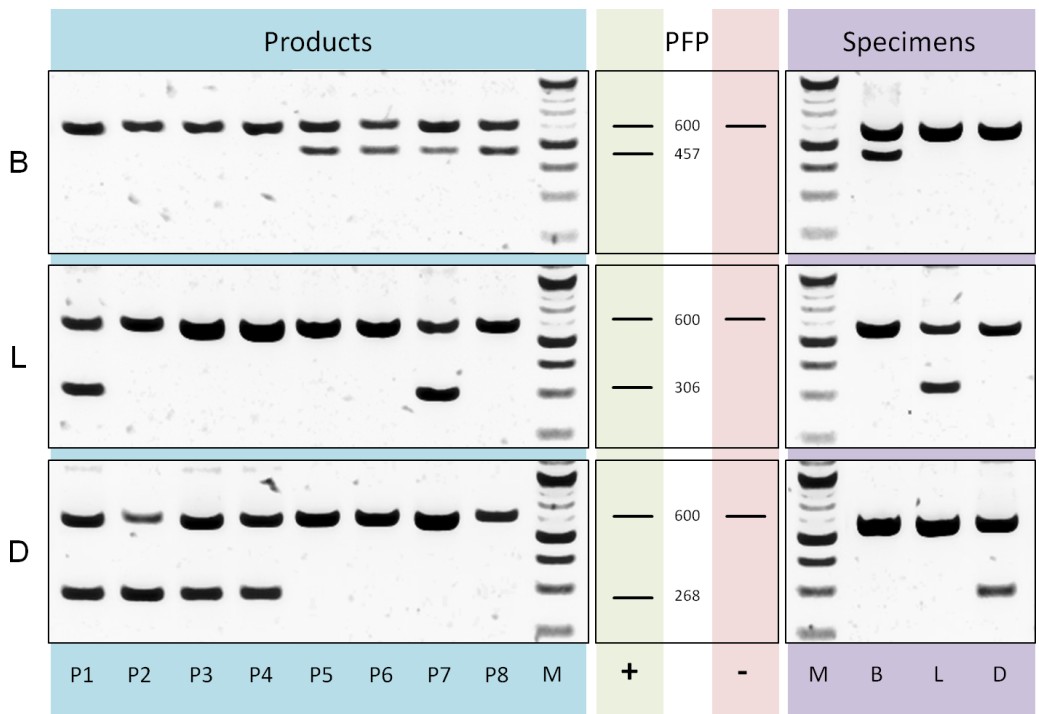

**Figure 5** **PCR diagnostics.** Comparison of multiplex PCR results using *rbcLa* universal primers and diagnostic (ARMS) primer. On the left are the results using DNA templates derived from products P1 to P8 with primer specific for bamboo (B), lemongrass (L) and *Dianthus* (D). Next, fragment pattern predictions (PFP) for presence (+) and absence (−) of corresponding components are depicted. The *rbcLa* fragment with a size of around 600 bp represents the positive reaction control. A smaller fragment is the diagnostic fragment and indicates the presence of a particular component (e.g., 306 bp fragment for lemongrass). On the right are representative results using DNA templates derived from bamboo (B), lemongrass (L) and *Dianthus* (D) specimens. For the approximation of fragment size a 100 bp (NEB) size standard (M) was used.

Comparing bambusoid genus diagnostics between all datasets (Fig. S1), the combined cytoplasmic dataset (Fig. 6A) provides the highest classification success. Using single locus *rbcLa*, only 14 of 23 bambusoid genera are at least partially diagnostically covered. In case of *rbcLb*, 20 of 23 genera are classified with three genera only partially (<50%) covered. The combination of *rbcLa* and *rbcLb* reflects the result of *rbcLb* with full coverage of two of these genera (*Fargesia* and *Pseudosasa*) and a slightly increased coverage of the third (*Phyllostachys*). Additionally, using provided LOGic formulas, the sequence of product sample P8 provides consistent characters (i.e., pos234 = T AND pos490 = T AND pos878 = G) with that of *Pseudosasa*. The diagnostic value of the *matK-KIM* region is similar to that of *rbcLa* with 13 of 23 genera at least partially covered. The combined dataset of *rbcL* and *matK-KIM* only leaves two genera without diagnostic pattern (i.e., *Semiarundinaria* and *Dendrocalamus*) and no false positives are detected (Fig. 6A). Using provided LOGic formulas, sequences of product samples P5–P7 contain consistent characters (i.e., pos12 = T AND pos263 = T AND pos701 ! = A AND pos738 ! = C AND pos1434 = G) with that of *Phyllostachys*.

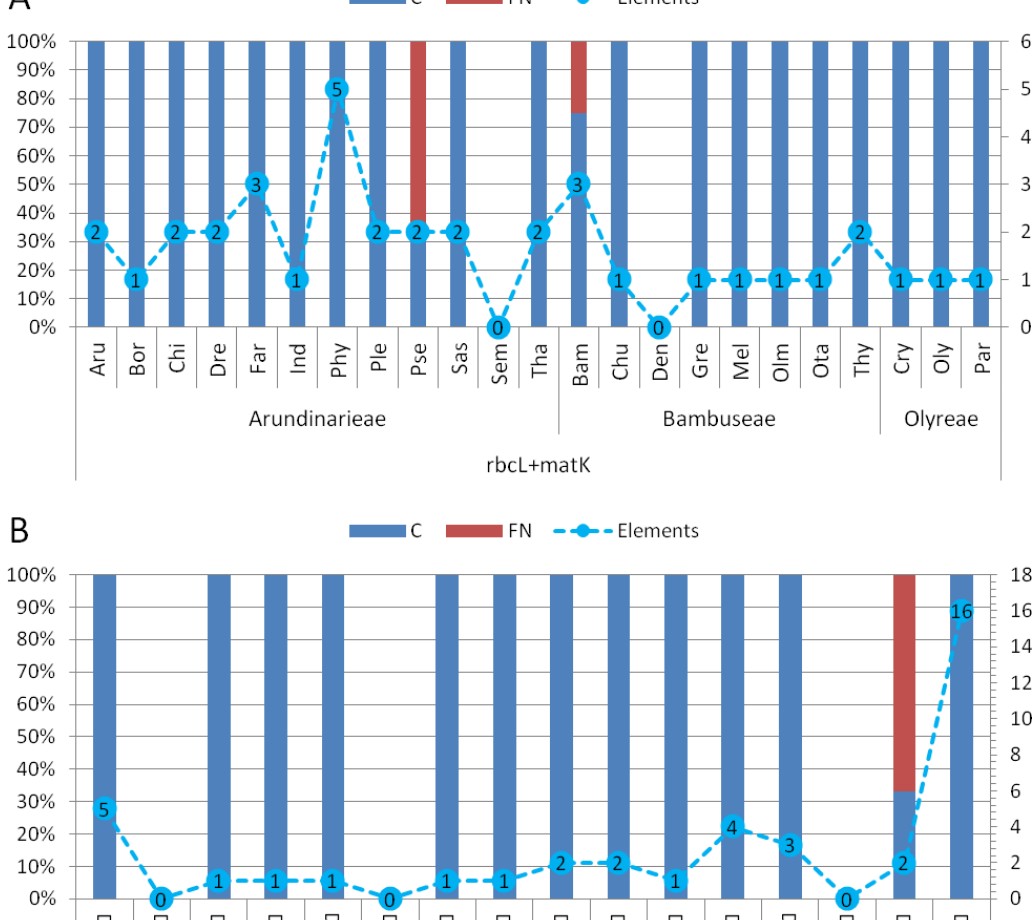

**Figure 6 Character based DNA diagnostics.** Barcoding with LOGic formulas (BLOG) analysis of bamboo genera (A) using the combined cytoplasmic dataset (*rbcLa*, *rbcLb* and *matK-KIM*), and *Dianthus* (B) using *matK-KIM* and *ITS* (extended) single marker datasets. Results for *Dianthus* species that only were present in the extended *ITS* dataset are not shown. Names of bamboo genera and *Dianthus* species are abbreviated with the first three letters of the corresponding genus and species name respectively (Table 2 and Table S2). The proportion (in %, primary *y*-axis) of coverage (C, blue) and false negatives (FN, red) using logic formulas is shown as bars. The number of elements (nucleotide positions) within the LOGic formula are represented by a dashed line graph (secondary *y*-axis).

All markers, either as single or in combination, offer diagnostic solutions for the genera of *Olyreae*. While *Arundinarieae* genera are moderately covered using *rbcLa* data and are almost completely void of diagnostic solutions considering *matK-KIM* data, in *Bambuseae* the situation is reversed, *matK-KIM* being more informative. Regarding single locus diagnostics, *rbcLb* is the superior region in the bambusoid group.

Comparing *matK-KIM* and *ITS* datasets for *Dianthus* (Fig. 6B) shows the inability to distinguish *D. chinensis* and *D. longicalyx* based on *matK-KIM* data. Using *ITS*, information

content increases enough to diagnose *D. chinensis* with a unique LOGic formula (pos181 = g AND pos595 = c) that also applies to product samples P1–P4.

## DISCUSSION

### Anatomical evaluations of commercial products

Due to the absence of bambusoid leaf characteristics in samples P1–P4, we can exclude a *Poaceae* and *Bambusoideae* origin of the leaves used in corresponding tea products. Stomata type and pattern of epidermal cells in comparison to specimens from the genus *Dianthus* suggest the origin of leaves to be found within this group. In contrast, observation of bambusoid leaf characteristics in samples P5–P8 leads to the conclusion that genuine bamboo leaves have been used in corresponding tea products. Investigating the possibility to differentiate between bambusoid tribes, the most promising feature appears to be tesselation. The ability to observe this pattern without or only limited magnification (<= 10×) in members of the *Arundinarieae* and the necessity of higher magnification (>= 40×) in members of the *Bambuseae* can be used to separate both "woody" bamboo tribes (*Farrelly, 1984*). Tesselation has also been observed with low magnification in samples P5–P8. This suggests that the source species for bamboo tea leaves are likely to be from the *Arundinarieae* tribe. Particular characteristics to differentiate between the bamboo genera were suggested by *Wu (1962)*. The wavyness of the walls of upper and lower epidermal cells in some species is different, while in other species the wavyness is constant. However, no quantification methodology nor any standard was suggested. Modifications of epidermal cells (i.e., uni- and bicellular hairs, spines, bristles and silica cells) also can contribute to a diagnostic evaluation but appear not to be exclusively distributed in one particular genus. Further studies are necessary to establish standards for potential diagnostic characters and to evaluate their phenotypic plasticity. One of the most challenging aspects of microscopic studies of dried bambusoid leaf samples are abundant papillae, often overarching the stomates (*Zhang & Clark, 2000*), and achieving sufficient clearing of the tissue samples. Tesselation is also a usefull diagnostic marker in separating bamboo from other *Poaceae* groups (e.g., lemongrass). Additional anatomical markers for this purpose are fusoid cells (*Motomura, Fujii & Suzuki, 2004*; *Ellis, 1988*; *Soderstrom & Ellis, 1988*) and invaginated arm cells in the chlorenchyma (*Zhang & Clark, 2000*). Both cell types, however, only can be observed in cross sections. Due to the processed nature (i.e., drying) of product samples, a more laborious sample preparation method (embedding) would be required to access these markers and results are likely to be biased by artefacts introduced by the drying process (e.g., collapsed parenchymatic cells). Based on our analyses, we compiled an anatomic diagnostics key for the differentiation of bamboo, lemongrass and carnation using markers available after a simple preparation step (Table S5).

### DNA based evaluations

Using DNA sequence data, we answer the question of the taxonomic origin of bamboo leaves used in commercial products. Since there are currently no DNA sequence markers that offer unambiguous data of bamboo diversity, thus unambiguous diagnostic signals, we cannot determine the specific origin. Not rarely, science tries to answer questions against

the odds by pragmatically modifying a question. Although we would be interested in exactly knowing the species names of bamboo tea leaves, to improve the current situation and to increase consumer safety, it is sufficient to answer modified questions. On the one hand we have the NFC, containing obscure and vague information about bamboo as source for tea leaves, associating it with a taxon from the tropical bamboos (*Bambuseae*). On the other hand, there is literature that suggests the origin to be from the temperate bamboos (*Arundinarieae*). We therefore ask: Which of the two groups are the leaves from? Are there reliable subordinate taxonomic units that help to locate the origin even further?

**Phylogenetic analyses**: Recent studies used Basic Local Alignment Search Tool (BLAST) in context of DNA barcoding to determine the identity of samples and to demonstrate the usefulness of DNA sequence data for the identification of food samples (*Stoeckle et al., 2011*; *Newmaster et al., 2013*). However, it has been only vaguely pointed out, that in context of DNA barcoding, which either aims to establish species barcodes by evaluating sequence data or to determine the species name of a specimen by using these barcodes, BLAST should not be mistaken for an accurate and consistent species identification approach (*Little, 2011*).

We need to be aware about the difference between identification and differentiation. While it is almost trivial to differentiate species from different genera, even with quite limited markers, it is next to impossible to unambiguously identify a species with the same marker. Basic requirements (e.g., sequence information from well documented and correctly identified specimens) of a limited set of only distantly related taxonomic entities are much more easily satisfied than those of an entire taxon. Although BLAST can be used to approximate identity, it does not offer sufficient data to be used as an identification facility on any taxonomic level. A hit with 100% sequence identity is not sufficient to determine taxonomic identity nor is the absence sufficient to determine that the sequence was derived from an entity that is not part of the corresponding taxonomic unit. BLAST does not tell us how many species of a genus are included in the database and does not know or tell anything about the resolution power of the used genetic region. Consequently, sufficient taxonomic coverage and resolution of sequence data has to be evaluated before any identification approach. Particularly, information of the taxonomic unit in question and closely related units are of critical importance. In situations where taxonomic coverage is patchy, phylogenetic approaches offer alternative solutions with measures of certainty about taxonomic units and are to be preferred over a simple BLAST result. To answer the questions formulated above, we used several phylogenetic models to reliably determine taxonomic units that include DNA sequences of commercial bamboo leaf samples.

Although we failed to include ITS, combining *rbcL* and *matK* sequence data in a phylogenetic analysis, enabled us to limit the possible taxonomic origin of bamboo tea leaves to the temperate bambusoid tribe *Arundinarieae*. Additionally, three out of four genuine bamboo product samples could be traced to a moderately supported subordinate clade (*Phyllostachys*, Fig. 3-6). Although morphological traits used to determine the genus of bamboo specimens were shown to be highly congruent with plastid RFLP data and the plastid genome has been extensively evaluated for its phylogenetic and phylogenomic potential (*Watanabe, Ito & Kurita, 1994*), information recovered from *rbcL* and *matK* was

not sufficient to resolve diversification among temperate "woody" clades (*Arundinarieae*). Our data confirms results of the analysis of six bamboo chloroplast genomes, that revealed low levels of variation in *Bambusoideae Zhang, Ma & Li (2011)*. The dataset was sufficient to unambiguously and securely determine that four of eight commercial samples belonged to a non-bamboo clade—*Dianthus* (*Caryophyllaceae*). Additionally several clades within *Dianthus* are strongly supported and three of the eight species form monophyletic clades.

The problems we encountered in our attempt to retrieve *ITS* sequences from bamboo specimens can be overcome by either cloning PCR products or using next generation instead of Sanger sequencing technology—a so-called metabarcoding approach (*Fahner et al., 2016*; *Hajibabaei et al., 2016*). Both strategies are either more laborious or more cost intensive but can handle the presence of multiple PCR fragment populations, a common situation when using the universal *ITS* marker or when the DNA template contains different genomes. In our study we applied a phylogenetic approach to use genomic regions that have been shown to be inefficient in tracing species of specimens (*Cai et al., 2012*), but are adequate for identifying higher level taxonomic assemblies. For species level identification more information is necessary, either in the form of new, more variable markers or by using technologies that can extract more information from already available markers. While *ITS* has been used in bamboo, currently available data is taxonomically patchy at best. Broader sampling among and within bamboo genera is necessary and could be complemented by deep sequencing (NGS) approaches. Fungal contamination and *ITS* paralogs that were considered problematic before, then would deliver additional information with potential for diagnostic solutions. The current situation, however, demands solutions that can be used immediately by companies and authorities to detect adulterations and protect consumers.

**Basic diagnostic assay**: Utilising the highly universal marker *rbcLa*, we introduced a PCR based diagnostic solution to detect DNA from bamboo (*Bambusoideae*), lemongrass (*Cymbopogon*) and carnation (*Dianthus*) species. By this, it can be determined if a product contains DNA from a genuine bamboo species, from another *Poaceae* species and from the potential adulteration genus of carnations. In the current rather unclear legal situation, this solution offers a basis for immediate action.

Results of the phylogenetic analyses and the content of literature about bamboo tea support the assumption that bamboo tea leaves are derived only from few different species. Advanced probes could be designed to detect and identify DNA from certain bambusoid genera or even particular species. Consequently, we evaluated the available sequence data and pinpointed positions of diagnostic value.

**Extended diagnostic assessments**: Using the combined cytoplasmic dataset in a character based approach (i.e., BLOG), we were able to trace characteristic patterns (LOGic formulas) of two bambusoid genera (i.e., *Phyllostachys* and *Pseudosasa*) to the genuine bamboo product samples (P5–P7 and P8 respectively). Despite the limitations of sequence based approaches, the character based approach demonstrates the diagnostic value of *rbcL* and *matK* on the generic level in bamboos and provides solutions to diagnose most (19 of 23) of the bamboo genera for which *rbcL* and *matK* sequence information is currently available in GenBank. However, the transfer of characteristic nucleotide patterns to a probe based diagnostic assay is not trivial, particularly in the case when different

markers are used in combination (i.e., *rbcL* and *matK*). The pattern recognized for the genus *Phyllostachys*, for example, requires the simultaneous detection of five nucleotide positions, thus the combination of five probes with four standard marker primers in a multiplex PCR. Additionally, when considering situations where the diagnostic nucleotides are located within close proximity of each other, we are confronted with the limitation of quick and cheap agarose based separation methods. More laborious and expensive methods (e.g., capillary electrophoresis of fluorescently labelled fragments) would have to be applied to overcome these limitations.

While using *rbcL* data only one of eight *Dianthus* species could be distinguished (data not shown) and no unambiguous pattern could be connected to the product samples considering *matK* data, using 85 *ITS* sequences retrieved from specimens and GenBank, the characteristic pattern of *Dianthus chinensis* could be traced to the samples taken from adulterated products (Fig. 6). Similar to the bamboo dataset, transfer of characteristic nucleotide patterns to a probe based diagnostic assay not always is trivial. Only one species can be diagnosed using one probe (*D. deltoides*), five need up to four probes and one even 16 (*D. turkestanicus*). The adulterant species *D. chinensis* can be diagnosed using two probes in combination with *ITS* standard marker primers.

## Conclusion

**Legal scientific framework**: Article 2 of the European General Food Law Regulation (*European Commission, 2002*) specifies "food" as any substance or product, whether processed, partially processed or unprocessed, intended to be, or reasonably expected to be ingested by humans. Tea products analysed in this study either consist of different "substances," one of which is "bamboo leaf," or only contain the latter. Consulting the List of Substances of the Competent Federal Government and Federal State Authorities (German version) for the category "plants and plant parts," common names (e.g., apple, lemon and orange) used in ingredient lists of teas are found and mapped to the scientific name of the corresponding plant the "substance" (e.g., fruit) is derived from. The common name "bamboo" can be mapped to two species of *Dendrocalamus* (*D. asper* and *D. latiflorus*) which are the source for bamboo sprouts. No other entries for bamboo are present. The English version of the mentioned list does not provide associations of common names with scientific names, representing one common unnecessary obstacle consumers and food business operators are confronted with—incoherent documentation of food related common and scientific names. Since bamboo is an exotic group, we have to assume that corresponding substances used in products fall under the novel food legislation and might be listed in the novel food catalogue.

Foods or food ingredients which have not been used for human consumption to a significant degree in the European Union (EU) before 15 May 1997 are governed by the provisions of the Novel Food Regulation (NFR) (*European Commission, 1997*). The Novel Food Catalogue (NFC) lists product components of animal and plant origin that are subject to the NFR or are being evaluated in that regard. The information is based on data provided by the EU member states. It is stated to be a non-exhaustive list and should serve as orientation on whether a product will need an authorisation under the NFR. Analysing the

content of the NFC, there are currently (June 2016) six species of four genera mentioned: *Bambusa oldhamii* (listed under the synonym: Sinocalamus oldhamii) *Dendrocalamus latiflorus*, *D. asper*, *Gigantochloa albociliata*, *G. levis* and *Phyllostachys edulis*. The immature shoot of these species is used as food substance and according to the NFC none of them are subject to the NFR. Additionally, there exists an entry for *Bambusa* species with a status indicating that history of use as a food of bamboo leaves is not known to any Member State and thus, bamboo leaves, if they were to be used as a food might be subject to the NFR and require a safety assessment before they may be placed on the market. According to this statement, based on current scientific data, the leaves of over 1,600 species of the *Bambusoideae* (*Poaceae*), if used as "substance" in tea, put corresponding products in violation of the NFR.

The same is most likely true for leaves of *Dianthus* species, particularly of the species *D. chinensis* which we found in tea products in place of genuine bamboo leaves. Due to their application in traditional Chinese medicine and contraindications for pregnant women, the admissibility as food has to be questioned.

Concluding, the use of the term bamboo for product components has several disadvantages. Firstly, a false impression of identity is promoted. Although the corresponding taxonomic entity has been shown to be monophyletic and offers unique characteristics, the contained morphological diversity deserves recognition beyond the subfamily rank. Secondly, the taxonomic range of the term may be perceived as ignorance and promote intentional adulteration or may lead to additional accidental confusions caused by lack of clarity. Any scientific approach for the safety assessment of botanicals and botanical preparations needs precision in regard of the corresponding taxonomy. Using terms like bamboo will always proof to be negligent and impede precise diagnostics. Additionally, if such a broad term would be used in context of the NFR, and corresponding products considered novel foods, the whole range of possible source species had to be included in safety assessments. This would require unnecessary high investments for the marketing of products that in reality only use few of these species as source. Experience tells us, that we cannot identify all natural units with little effort. In order to be able to differentiate on a level where genetic markers show coherence between the unit and its inherited chemical profiles—which ultimately is the empirical dimension used to assess safety—systematic knowledge is of primary importance.

**What is bamboo tea?:** There are obviously different opinions or assumptions to the question which taxonomic unit actually can be regarded as the source for bamboo tea leaves. The NFC of the EU assumes that leaves of the tropical "woody" genus *Bambusa* (*Bambuseae*) might be used. Literature suggests that species of temperate "woody" genera (*Arundinarieae*) have been used, and thus might still be used as source for bamboo tea leaves. It has to be noted that taxonomy does not produce a completely static structure. New insights can lead to changes in the view of the tree of life. According to the NCBI Taxonomy the common name for the tribe *Bambuseae* is bamboo. This reflects an old systematic opinion (*Zhang & Clark, 2000*) when *Bambuseae* still contained most *Arundinarieae* genera (e.g., *Sasa* and *Phyllostachys*) and may be the reason why *Bambusa* has been associate with bamboo leaves in the NFC.

The most recent scientific usage of the term bamboo is found in *Soreng et al. (2015)* where bamboo is the common name for the subfamily *Bambusoideae* (*Poaceae*). This group is characterized by high morphological diversity that appears not to be discretely associated with subordinate taxonomic entities. The reasons are believed to be related to morphological inter-gradation interpreted in various ways and the presence of hybrids that have been stabilized through clonal propagation (*Triplett & Clark, 2010*). The taxonomic confusion within the group also is related to a peculiarity of the reproduction mode of bamboo. While most flowering plants are flowering regularly each year, bamboo is one of the groups where dramatically extended intervals exist—some as long as 120 years (*Veller, Nowak & Davis, 2015*; *Liese, 2015*).

Although DNA based approaches to classification of bamboos are confronted with limited resolution of genetic markers, the subfamily has been well established and the temperate "woody" clade (*Arundinarieae*) was resolved to an acceptable degree, delivering additional information about associations of particular genera and biogeographic hypotheses (*Triplett & Clark, 2010*). All genuine bamboo leaves analysed in the present study could be placed within the *Arundinarieae* tribe using macroscopic leaf characteristics. Furthermore, they could be traced to internal groups by phylogenetic methodology (*Phyllostachys* clade) and a character based DNA barcoding approach (*Phyllostachys* and *Pseudosasa* genera).

**Carnation = bamboo tea?:** From an evolutionary perspective, bamboo and carnation are fairly different groups of plants with more than a hundred million years of independent development (*Chaw et al., 2004*). How is it possible to confuse such distinct groups? Scientific names exist because they allow us to communicate precisely. However, since it is also common for humans to label things by its appearance, it is not surprising to find a simple explanation for a potentially severe adulteration of teas supposedly containing bamboo leaves:

A product description (retrieved in July 2014) of so called "bamboo tea carnation" is advertised by the following sentence: "There are well over a hundred varieties of bamboo growing in China. This is not one of them, actually belonging to the genus of Carnations (*Dianthus*), but the young shoots closely resemble bamboo in appearance..."

Communication using the term bamboo in conjunction with tea obviously is ambiguous and may have caused the adulteration of products we analysed. Since these products had been on the marked for at least 1.5 years before they were discontinued, we must ask what consequences this may have had for consumers? Several species of carnation are mentioned in an ethno-medicinal context (*Chandra & Rawat, 2015*). Particularly in traditional Chinese medicine two species—*D. chinensis* and *D. superbus*—are widely used as Dianthi herba for the treatment of diuresis and strangury (*Chinese Pharmacopoeia Commission, 2010*). Chemical constituents are saponins (*Oshima, Ohsawa & Hikino, 1984*; *Hong-Yu, Koike & Ohmoto, 1994*), flavonoids, sterol, glycosides and cyclopeptides (*Han et al., 2015*; *Han et al., 2014*; *Hsieh et al., 2004*). Studies on bioactivity have shown various effects. Cyclopeptides for example showed anti-bacterial, anti-fungal, estrogen-like, uterotonic, haemolytic and cardio-toxic effects. The uterotonic effect is the reason why Qu mai (Dianthi herba) should not be prescribed to pregnant women (*Wu, 2005*). By selling tea that lists bamboo leaves

as ingredient but actually contains leaves of *Dianthus* species, consumers are mislead. Additionally, if the *Dianthus* species is known to have an effect on the dynamics of the uterus, pregnant women are put in harms way. Our data strongly suggests that leaves found instead of bamboo leaves are from *D. chinensis* and measures to prevent this kind of misdirection have to be implemented immediately.

## ACKNOWLEDGEMENTS

We applied the SDC approach for the sequence of authors (*Tscharntke et al., 2007*). We thank the garden staff of the Karlsruhe Institute of Technology for taking good care of the specimens and the trainees of the botanical institute for their contributions to this work. Lastly, we acknowledge the contributions of Esther Huber with her bachelor's thesis (*Huber, 2014*) about bamboo and its use as food.

### Funding

We received support by Deutsche Forschungsgemeinschaft and Open Access Publishing Fund of Karlsruhe Institute of Technology. The funders had no role in study design, data collection and analysis, decision to publish, or preparation of the manuscript.

### Grant Disclosures

The following grant information was disclosed by the authors:
Karlsruhe Institute of Technology.

### Competing Interests

The authors declare there are no competing interests.

### Author Contributions

- Thomas Horn conceived and designed the experiments, performed the experiments, analyzed the data, wrote the paper, prepared figures and/or tables, reviewed drafts of the paper, and performed the morphological specimen determination (*Dianthus*).
- Annette Häser performed the experiments, reviewed drafts of the paper, and performed the morphological specimen determination (Bamboo & *Dianthus*).

### DNA Deposition

The following information was supplied regarding the deposition of DNA sequences:
GenBank
KU722849–KU722908
KU748524–KU748525
KX146413–KX146425
KX146427–KX146439
KX146450–KX146462
KX233492–KX233495
KX233501–KX233508.

## Data Availability

TreeBase: http://purl.org/phylo/treebase/phylows/study/TB2:S19113.

## Supplemental Information

Supplemental information for this article can be found online at http://dx.doi.org/10.7717/peerj.2781#supplemental-information.

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
