# Peer review of "Bamboo tea: reduction of taxonomic complexity and application of DNA diagnostics based on rbcL and matK sequence data"

_PeerJ, doi:10.7717/peerj.2781_

## Round 0.1 · original submission · Major Revisions

All three reviewers have several concerns and suggestions to improve the manuscript. Please, revise the manuscript according to the suggestions of the reviewers.

Reviewer 1 ·

Basic reporting

The paper entitled "Adulteration of herbal products: bamboo tea authentication" identifies whether tea products indeed contain bamboo species as advertised by means of DNA barcoding and by anatomical observations. The objectives are not clearly stated and the paper is not clearly written in most of the sections as well. Moreover by using two DNA chloroplast markers previous research identified more species involved in teas or herbal infusions than in this paper and by using only BLAST, without performing phylogenetic and phenetic methods.

Introduction lacks the most important references for DNA barcoding for plants (e.g. Hollingsworth 2011) and does not explain that CBOL suggested two main chloroplast markers and two additional regions to identify plant species. They used ITS a nuclear marker without clearly explaining. On the other hand, Introduction lacks important references published specifically to identify components in teas and herbal infusions such like Soeckle et al. 2011 (Nature) or Galimberti et al. 2013 and Wallace et al. 2012 (Food Research International). There is a plethora of papers dealing with identification of food products using DNA barcoding and the manuscript does not address this issue in the short and incomplete introduction. Furthermore it is not important to include the taxonomic section explaining bamboo classification and diversification and only explain which species are used in teas.

Experimental design

Material and Methods starts explaining that they cultivated several bamboo species, Cymbopogon, the lemmon grass and Dianthus, the carnation. It is not clear why they cultivated these non-bamboo species and later they explain that they found them in the bamboo teas according to the DNA barcoding. This section does not specify which teas where analyzed, it is only mentioned that they acquired the teas in a German internet site and also some local teas without more information. They identified the components of these teas by anatomical observations comparing the samples with the plants they cultivated and by DNA sequences (nuclear ITS, and plastid matK and rbcL). They were not able to sequence ITS for all samples. They performed several methods such like ML, Parsimony and UPGMA to assess whether samples either belonged to the same clade or were grouped in the same cluster. Most recent papers dealing with identification of species in food products use only BLAST, demonstrating that with this facility it is possible to easily identify to species level or at least to genus. Moreover, it was not necessary to include the anatomical observations.

Validity of the findings

It has been amply demonstrated that solely with DNA barcoding is possible to identify food elements without going into so much trouble of getting anatomical characters. In this paper authors had to identify first which species were involved in the herbal teas by DNA barcoding to get later anatomical characters. Moreover, all DNA barcode papers were able to identify all components in tea or in other food products by BLAST without more analyses. Stoeckle et al. (2011) were able to identify more than 100 species by sequencing rbcL and matK in teas and herbal teas, and that the success of the ID was in relation with having closely related species in GenBank. Therefore, without cultivating the bamboos and non-bamboo species only by using the DNA markers is possible to identify not only bamboos, lemon grass and carnation but other components in teas.

Additional comments

The paper will improve by including the most important references of previous DNA barcoding references to identify components in teas and in other food products and by precisely explaining the usefulness of DNA barcode for this purpose. In methods there is no need to obtain anatomical characters to identify species nor cultivate plants. Moreover, only by using BLAST in GenBank is possible to identify species without performing more phylogenetic and phenetic analyses and percentage of identification should be included.

Reviewer 2 ·

Basic reporting

The market survey was not very clear to me and it requires more evidence on how it’s been done. Additionally, this section needs to be stretched out a bit more.

Experimental design

none

Validity of the findings

none

Additional comments

COMMENTS FOR THE AUTHOR:
MS title: Adulteration of herbal products: Bamboo tea authentication (#11598)
Authors: Horn and Haser
The manuscript by Horn and Haser, describes the DNA analysis of a group of bamboo tea products that were manufactured in Germany in order to determine the bamboo and related plants parts of adulteration, substitution and contamination within the market. This area of research is extremely important for the safety of consumers and also to law enforcement where endangered species or illegal ingredients are concerned. As the authors describe, DNA barcoding is an excellent method to utilise for authentication of bamboo tea, and is more accurate for species level identifications over traditional analytical chemistry methods. The building of a bamboo Species Reference Material barcode library is a fantastic and much needed resource for authenticating bamboo tea, especially with the use of three gene regions (rbcL, matK and ITS2) in combination, as has been done by the authors in this manuscript. It is definitely agreed that DNA-based methods should be incorporated into authentication tools for assessing bamboo tea products, a key point made by the authors.
There are few suggestions below that mainly relate to the methods, and possibly the discussion, section for consideration by the authors, as minor revisions to the manuscript.
These are:
1) The technique is a quite mature method with the authors widely practiced for a long time and hence lack of novelty. Only the workload with massive samples is the highlight if any.
2) Can you provide more detail on the methodology behind the analysis of the bamboo product market survey & sequence data? What programs were used to align the sample sequences with the species reference material sequences? Were percentage identities used to determine species/genus/family level identifications, or tree based methods? Were any extraction/negative controls used?
3) Consider whether it is useful to carry out a BLAST search of the GenBank database with your sample sequences as well as against the bamboo Species Reference material database. Are there reference sequences already available for more species than were included in the bamboo Species Reference material database that could provide species level resolution where this wasn't possible with the bamboo Species Reference material?
4) The use of Sanger sequencing is not the best method to use for identifying multiple ingredients in a bamboo products mixtures. Even with a number of replicates, it may not be possible to detect every ingredient in a mixed sample due to different levels of DNA preservation and primer bias for one taxa over another. A metabarcoding approach is more appropriate for screening samples that have multiple ingredients. A suggestion for the authors is to mention the application of next generation sequencing to analysing bamboo mixtures – tea samples.
5) What is the criteria for a sample to be authentic or not authentic in your study?
6) What is the significance of adulterants? are they natural substitutes?
7) There is a difference between adulterant and substitute. The authors need to clarify which samples were adulterated with an unwanted species (adulterant; mixed sample) or if they and which were actually replaced with an unwanted species (substitute).
8) Adulterant samples: did the authors always find 'clean' sequence traces or also mixed sequence traces, which would suggest a mixed tea samples? "Our intensive sampling yielded a maximum of 2-3 species barcodes per tea product" did you adopt clone technique"
9) Did the authors try to amplify the psbA-trnH sequence as well when the ITS2 sequence failed to amplify? Did the authors tried tired approach?
10) Authors need to provide more detail:
a) from how many samples did the DNA extraction fail? a Was it repeated, was a different method applied to extract DNA?
b) from how many samples did the PCR amplification fail? a Was it repeated? Were different DNA concentration tested? The absence of PCR products can have several reasons.
c) from how many samples did the sequencing fail? What do the authors consider as failure? Low quality sequences or mixed sequence traces? The later would be indicating the sample contains mixed DNA (= adulterated, not substituted)
d) Plant materials. What was used as voucher/reference samples to provide the species specific sequences? Who identified the voucher samples?

Finally, The below-mentioned papers are examples to synthesize more information to the manuscript.
• The August 2014 issue of Forensic Science International had published review article by S.K Verma, entitled 'DNA Evidence: Current Perspective and Future Challenges in India'. http://authors.elsevier.com/a/1PMrL1MCG01les
• Seethapathy, G.S., D. Ganesh, J. U. Santhosh Kumar, U. Senthilkumar, S.G. Newmaster, S. Ragupathy, R. Uma Shaanker and G. Ravikanth. 2014. Assessing product adulteration in natural health products for laxative yielding plants, Cassia, Senna, and Chamaecrista, in Southern India using DNA barcoding. International Journal of Legal Medicine. DOI 10.1007/s00414-014-1120-z
• G. Ferri1, , ,; B. Corradini1,; F. Ferrari,; A.L. ... www.sciencedirect.com/science/article/pii/S187249731400221X by G Ferri - ‎2014Oct 16, 2014 - Forensic botany II, DNA barcode for land plants: Which markers after the international agreement?
• Nithaniyal S, Newmaster SG, Ragupathy S, Krishnamoorthy D, Vassou SL, et al. (2014) DNA Barcode Authentication of Wood Samples of Threatened and Commercial Timber Trees within the Tropical Dry Evergreen Forest of India. PLoS ONE 9(9): e107669. doi:10.1371/journal.pone.0107669
• Purushothaman, N. S.G. Newmaster, S. Ragupathy, N. Stalin, D. Suresh, D.R. Arunraj, G. Gnanasekaran, S.L. Vassou, D. Narasimhan and M. Parani. 2014. A tiered barcode authentication tool to differentiate medicinal Cassia species in India. Genetics and Molecular Research 13 (2): 2959-2968.
• Mahadani, P. and Ghosh, S.K. (2013) DNA Barcoding: A tool for species identification fromherbal juices. DNA Barcode, 1, 35-38.
• Revathy, S.S., Rathinamala, R. and Murugesan, M. (2012) Authentication methods for drugs used in Ayurveda, Siddha and Unani systems of medicine: An overview. Int.J.Pharm. Sci. Res. 3, 2352-2361
• Stoeckle, M.Y., Gamble, C.C., Kirpekar, R., Young, G., Ahmed, S. and Little, D.P.Commercial teas highlight plant DNA barcode identification successes and obstacles.Sci. Rep. (2011) 1, 42. doi:10.1038/srep00042
• Wallace, L.J., Boilard, S.M.A.L., Eagle, S.H.C., Spall, J.L., Shokaralla, S. and Hajibabaei. M.(2012) DNA barcodes for everyday life : routine authentication of natural healthproducts. Food. Res. Int. 49, 446-452.

Reviewer 3 ·

Basic reporting

 Introduction section needs to be re-organised with proper background concerning the details for “Identification of Herbal Product Components” part. It could be helpful to have more explanation/definition of the alternative methods for plant authentication and why they aren’t as successful or useful as molecular methods.
 Gene names should be italicized throughout the manuscript.
 GenBank sequence accessions mentioned in entire manuscript (Table 1, 2 etc) doesn’t exist in GenBank record.
 DNA Barcoding should be mentioned correctly as DNA barcoding.
 Genbank should be mentioned correctly as GenBank.
 The specimens used in the study have been mentioned to be acquired and cultivated in the botanical garden of the Karlsruhe Institute of Technology. However no voucher details have been reported in the manuscript.
 The DNA extractions, PCR and sequencing protocols mentioned in the manuscript were standard but too brief in context. Authors have not clarified the standard pcr amplification programme used for any of the tested primers. However, it needs to be more detailed.
 Authors have mentioned the use of two rbcL primers: rbcLa and rbcLb. What is the utility of testing two universal primers from the single region (rbcL gene) either in alone or in combination on a single dataset?
 Authors have claim (Page 2, Line No. 80-81) that combination of official barcode rbcL+matK has been shown to be of limited use for species-level specimen identification in land plants. On the contrary, during their study they had tested the same combination only. Why the combinations of chloroplast and nuclear region primer viz. rbcL+ITS or matK+ITS have not been tried for?
 Nuclear region ITS have been employed only in context of genus Dianthus. What is the probable reason behind the same? Why it has not been tested in bamboo and lemongrass datasets inspite of 100% PCR success reported there in.
 The authors evaluate the efficacy of DNA barcoding to resolve the taxonomic origin of genuine bamboo leaves used in commercial products. However the methodology used on the concept of DNA barcoding is not to the point based on the published literatures in the past. In terms of DNA based evaluation, authors had only used the phylogenetic analysis to conclude the results. However, the phylogenetic analyses do apply appropriate models, and the outcomes of parsimony and likelihood trees are reported. But the complete ignorance of distance and similarity based approaches to conclude the potentiality of DNA barcoding markers through the use of DNA barcoding gap which are in concern with the authorisation and use of novel foods and food ingredients related to consumer health and safety should be avoided.
 The authors had recommended precise conclusion through their studies. Overall the work is quite genuine and relevant in context to herbal market. The results presented are interesting. Limited number of tested regions and methodologies present a logic concern to the output of the study. Much of the interpretations are slightly confusing. But there are certain approaches which had been completely overlooked during the drafting of manuscript. Authors should discuss emerging approach of DNA barcoding (High Resolution Melting Curve Analysis) as a part of introduction which focus the application of DNA barcoding combined with high resolution melting (Bar- HRM) analysis as a novel, advanced method which has recently been successfully applied in herbal medicine authentication. This analysis is widely used in many researches for contamination detection in herbal mixture. Also the potential of NGS for metabarcoding of herbal supplements should be discussed there in. The successful literatures published till date provides evidence based support to the technology and should be referred to during the study design.
 Figure 3 seems unclear in interpreting the concept behind. Author should describe the proper background of the concept used for the same.

Experimental design

Page 1, Line No. 40: The statement “Which kind of bamboo are the leaves taken from that are used in bamboo tea?” is grammatically incorrect. It should be: Which kinds of bamboo are the leaves taken from, that are used in bamboo tea
Page 2, Line No. 49: Should be For ages,
Page 2, Line No. 50: bamboo growing/ producing countries (few countries name can be specified)
Page 2, Line No. 55: Should be In China and Indonesia,
Page 2, Line No. 69: The sentence “The classic approach to identify herbal product components is based on described anatomical features of involved plant parts” seems to be hypothetical. The term “classic approach to identify herbal product components” in general does not only limit to anatomical features. Or if the author is trying to focus the specific way of identification prevailing there in terms of Bamboo or herbal product authentication, should be presented with specific reference accordingly.
Page 2, Line No. 78: Should be verified species reference database....
Page 2, Line No. 81-82: gene name should be italicized.
Page 2, Line No. 82-84: The statements are ambiguous which needs to be refined in terms of specificity.
Page 3, Line No.107-108: The link reported for the specimen details should be specified in context to study. Avoid mentioning the common link.
Page 3, Line No.108: Commercial products were acquired from local and internet sources (Table 2). No details for the commercial products have been reported clearly in the table. What does the code P1 to P8 stands for? The acquisition of commercial products from the internet sources seems to be vague. Author should clear the ambiguity.
Page 3, Line No.118: For DNA based evaluations; we chose to retrieve sequence information from...
Page 3, Line No.126: lemongrass (check the spelling consistency throughout the manuscript).
Page 10, Line No.324: positive
Page 10, Line No.325: negative

Validity of the findings

No Comments

---

## Round 0.2 · Major Revisions

The revised manuscript has improved. However, reviewer 1 still has major concerns about the revised mansucript. Please, try to address the concerns of reviewer 1 and implement the required improvements in a second revision. Rather than placing comments within the manuscript, provide a letter indicating the revsions within the mansucript.

Reviewer 1 ·

Basic reporting

The R1 version of the manuscript was greatly improved, however I find that authors do not follow several suggestions, for example the revison by a native English speaker.
In addition they do not use standard English for certain terms, e.g. citing spp. I found ortographic errors such like "lemongras" I find as well that the rebuttal letter is difficult to follow because the word file by the reviewers was used and comments were added to this file.
In the introduction the most important references for DNA barcode for plants are still lacking. Moreover they do not cite references for the classification of Poaceae in particular for bamboos, they use a loose concept of "bamboo" in Introduction but in results they use taxonomic categories that were not explained before. Scientific names authors were not included.

Experimental design

They confuse phylogenetic analyses with phenetics, they considered that UPGMA is a phylogenetic method! They report results like substitution models that were not included in methods. Authors use "matK-KIM" because they used the primers designed by KIM, and is totally incorrect. They obtained ITS sequences and they did not include them in analyses.

Anatomical figures do not have enough quality to be published in PeerJ.

Validity of the findings

I found that with all these flaws their findings can not be corroborated.

Additional comments

In this R1 authors failed to adress the most important issues raised before and their rebuttal letter is difficult to follow. The most important references for DNA barcoding in plants are still lacking, classification of Bambusoideae was not included and it is used in results. Introduction do not stress the importance of the barcode for identifiyng species included in food products. Authors continue using methods such like UPGMA as phylogenetic methods. Results continue using anatomical observations in figures lacking qualitiy. With all these flaws my suggestion is to reject the paper.

Reviewer 3 ·

Basic reporting

The authors have taken care to revise the manuscript according to the comments. The questions/hypotheses are well defined.

Experimental design

The experimental approach is adequately described and the motivation for each experiment is provided.

Validity of the findings

The data/findings are clearly described.

Additional comments

The authors have taken effort to address all the comments critically. The manuscript can be accepted for publication in the present form.

---

## Round 0.3 · accepted · Accept

After the reviewer 2 did not respond to an invitation to re-review, and after reviewing the materials, I have decided for an ACCEPT decision.